# Superior Survival and Lower Recurrence Outcomes with Breast-Conserving Surgery Compared to Mastectomy Following Neoadjuvant Therapy in 607 Breast Cancer Patients

**DOI:** 10.3390/cancers17050766

**Published:** 2025-02-24

**Authors:** Damiano Gentile, Jacopo Canzian, Erika Barbieri, Simone Di Maria Grimaldi, Rita De Sanctis, Corrado Tinterri

**Affiliations:** 1Breast Unit, IRCCS Humanitas Research Hospital, Via Manzoni 56, Rozzano, 20089 Milan, Italy; erika.barbieri@cancercenter.humanitas.it (E.B.); simone.dimariagrimaldi@cancercenter.humanitas.it (S.D.M.G.); corrado.tinterri@hunimed.eu (C.T.); 2Department of Biomedical Sciences, Humanitas University, Via Rita Levi Montalcini 4, Pieve Emanuele, 20090 Milan, Italy; jacopo.canzian@cancercenter.humanitas.it (J.C.); rita.de_sanctis@hunimed.eu (R.D.S.); 3Medical Oncology and Hematology Unit, IRCCS Humanitas Research Hospital, Via Manzoni 56, Rozzano, 20089 Milan, Italy

**Keywords:** breast cancer, neoadjuvant therapy, breast-conserving surgery, mastectomy

## Abstract

This study investigates the impact of breast-conserving surgery (BCS) versus mastectomy on long-term outcomes in breast cancer (BC) patients treated with neoadjuvant therapy (NAT). A retrospective analysis of 607 patients treated at IRCCS Humanitas Research Hospital revealed that BCS, performed in 54.7% of cases, was associated with superior 10-year disease-free survival (DFS), distant DFS, overall survival (OS), and BC-specific survival (BCSS) compared to mastectomy. Pathologic complete response (pCR) significantly improved survival outcomes, while mastectomy was an independent predictor of worse BCSS. Multivariate analysis identified tumor stage, nodal status, and single-nodule presentation as predictors of BCS. These findings support the oncologic safety of BCS after NAT, emphasizing its advantages in appropriately selected patients. This study demonstrates the importance of individualized surgical decision-making to maximize survival while maintaining quality of life in BC patients undergoing NAT.

## 1. Introduction

Neoadjuvant therapy (NAT), defined as systemic treatment given before surgery, has become an important approach in the management of breast cancer (BC), especially in locally advanced BC and select cases of early-stage disease. Initially introduced in the 1970s to downstage inoperable tumors [1], NAT nowadays offers multiple benefits. It allows for the conversion of previously inoperable tumors into operable cases [2,3,4], improves surgical options by shrinking tumor burden, and enables breast-conserving surgery (BCS) in patients who might otherwise require mastectomy [5,6,7,8]. Additionally, NAT also allows for in vivo assessment of tumor biology and chemosensitivity, providing prognostic information and guiding subsequent adjuvant treatment strategies [9,10]. The prognostic significance of response to NAT has become increasingly evident, with pathologic complete response (pCR)—the absence of invasive cancer in both the breast and lymph nodes—being a strong predictor of long-term endpoints such as disease-free survival (DFS), distant DFS (DDFS), overall survival (OS), and BC-specific survival (BCSS) [9,11,12,13,14,15]. In addition, effective tumor downstaging by NAT has the potential to decrease the extent of surgery, decreasing the need for axillary lymph node dissection (ALND) and increasing the rates of BCS in patients initially scheduled for mastectomy [16,17,18,19]. Historically, BCS has been demonstrated to be an oncologically safe alternative to mastectomy in the upfront surgical setting, achieving comparable OS and DFS [20,21]. More recently, evidence suggests that BCS, also following NAT, may indeed confer superior survival compared with mastectomy [22,23,24,25,26]. This observation highlights the dual benefits of BCS, combining oncologic efficacy with improved cosmetic outcomes and quality of life [27,28,29]. Despite the well-documented advantages of NAT and the oncologic safety of BCS, its adoption following NAT remains inconsistent. This hesitancy can be attributed, in part, to conflicting evidence from large-scale studies comparing BCS and mastectomy in this context. For instance, a meta-analysis by the Early Breast Cancer Trialists’ Collaborative Group (EBCTCG) revealed that while NAT significantly increased the rate of BCS, it was associated with higher local recurrence rates compared to adjuvant chemotherapy [30]. This increase was most significant in patients treated with BCS, contributing to concerns about its oncologic adequacy in downsized tumors. Additionally, another meta-analysis encompassing over 5000 BC patients undergoing NAT demonstrated better 5-year DFS and OS rates for mastectomy compared with BCS [31]. However, both studies have notable limitations, such as heterogeneity in the selection of patients, tumor characteristics, and treatment protocols, as well as a lack of contemporary data to reflect advances in NAT regimens, imaging, and surgical techniques. The comparison of recurrence and survival outcomes between BCS and mastectomy in patients treated with NAT is therefore limited. Additionally, identifying independent prognostic factors influencing recurrence and survival in this population remains crucial for optimizing surgical decision-making. In the present study, we aimed to address these knowledge gaps by comparing the characteristics and long-term oncologic outcomes of patients undergoing BCS versus mastectomy after NAT. Furthermore, we sought to identify predictors of surgical treatment and independent prognostic factors associated with recurrence and survival.

## 2. Materials and Methods

This retrospective study analyzed all consecutive BC patients treated with NAT at the Breast Unit of IRCCS Humanitas Research Hospital, Milan, Italy, between October 2006 and December 2023. Patient data were extracted from an institutional database and medical records, including demographics, clinical and pathological tumor characteristics, imaging results, and treatment details. Key parameters collected were age, menopausal status, baseline tumor size, focality, clinical and pathological stage, nodal status, histotype, vascular invasion, and molecular subtype. Data on the type and number of systemic therapy cycles were also recorded.

### 2.1. Pre-Neoadjuvant Therapy Evaluation and Imaging

All patients underwent pre-NAT staging, which included bilateral breast and axillary ultrasound (US) and mammography. Magnetic resonance imaging (MRI) of the breast and/or total-body positron emission tomography (PET) scans were performed in most cases to assess tumor extension. The preoperative diagnosis of invasive BC was made by US-guided core needle biopsy (14-gauge), while axillary staging included biopsy of clinically or radiologically suspicious lymph nodes. Metallic clip markers were placed in the primary tumor under US guidance to facilitate tumor localization during surgery.

### 2.2. Neoadjuvant Therapy Regimens and Response Assessment

The indication for NAT, based on tumor size, clinical stage, and molecular subtype, was determined by the multidisciplinary tumor board comprising breast surgeons, breast medical oncologists, radiotherapists, breast radiologists, plastic surgeons, and pathologists. Response to systemic therapy was monitored by periodic physical examinations and imaging. Post-NAT, tumor size and response were assessed by US, MRI, or PET imaging, depending on the modalities used initially. pCR was defined as the absence of invasive or non-invasive cancer in both breast and axillary lymph node specimens.

### 2.3. Surgical Management

Following NAT, all patients underwent either BCS or mastectomy, with the decision tailored to tumor response. Specimen radiographs were obtained intraoperatively to confirm clip and lesion localization and assess margin status in BCS cases. Surgical specimens were inked and sectioned at 2- to 3 mm intervals for intraoperative pathological analysis. Axillary surgery consisted of either sentinel lymph node biopsy or direct ALND. Patients with macrometastatic sentinel lymph nodes on intraoperative assessment proceeded to ALND.

### 2.4. Pathological and Molecular Analysis

Surgical specimens were evaluated by two experienced breast pathologists. Tumor subtype and hormone receptor (HR) status were determined by immunohistochemistry (IHC), with estrogen receptor and progesterone receptor positivity defined as >1% immunoreactive cells [32]. HER2 status was evaluated using IHC and confirmed by fluorescence in situ hybridization (FISH) for IHC scores of 2+. HER2 positivity followed ASCO-CAP guidelines [33], with IHC 3+ or IHC 2+ with FISH amplification classified as positive. Tumors were categorized into subtypes (e.g., luminal-like, HER2-positive, and triple-negative) based on HR and HER2 status.

### 2.5. Inclusion and Exclusion Criteria

Inclusion criteria encompassed all patients with histologically confirmed BC who completed NAT and underwent surgery. Patients with metastatic disease at diagnosis, a history of previous BC or other malignancies, disease progression during NAT, or a follow-up of less than 12 months were excluded. Additionally, patients who received excisional biopsy or debulking surgery as their initial treatment were excluded. The flowchart is shown in Figure 1.

### 2.6. Adjuvant Therapy, Radiotherapy, and Follow-Up

Patients not achieving pCR after anthracycline- and taxane-based NAT were given adjuvant systemic therapies tailored to their tumor subtype, including capecitabine for triple-negative BC and trastuzumab emtansine (T-DM1) for HER2-positive BC. Adjuvant endocrine therapy was administered to all HR+ BC patients according to menopausal status: pre-menopausal patients received ovarian function suppression with an LH-RH analog plus exemestane, while post-menopausal patients were treated with aromatase inhibitors. All patients who underwent BCS following NAT received adjuvant whole-breast irradiation. The standard protocol included either hypofractionated radiotherapy (40.5 Gy to the whole breast with a tumor bed boost of 48 Gy, delivered in 15 fractions) or conventionally fractionated whole-breast irradiation (50 Gy in 25 fractions with an additional tumor bed boost of 10 Gy in 5 fractions). Post-mastectomy radiotherapy was recommended based on preoperative clinical staging and pathologic response after NAT. Post-mastectomy radiotherapy was routinely indicated for patients with an initial tumor size >5 cm or evidence of skin invasion (cT3-4), regardless of their tumor response to NAT. Additionally, adjuvant regional nodal irradiation was given to patients with cN+ disease at baseline who did not achieve a pathologic axillary response (ypN1a-2a-3) after NAT. The standard post-mastectomy radiotherapy dosing regimen consisted of 50–50.4 Gy in 1.8–2.0 Gy per fraction (25–28 total fractions) to the chest wall and 45–50 Gy in 1.8–2.0 Gy per fraction (25 total fractions) to the regional lymph nodes. Follow-up controls occurred every six months through routine physical examinations, laboratory analyses (e.g., carcinoma antigen 15-3 levels), and imaging studies with annual mammography, breast and axillary US, and chest X-rays. Written informed consent for surgery and data use was obtained from all patients.

### 2.7. Statistical Analysis

Descriptive statistics were summarized as frequencies and percentages for categorical variables or medians and ranges for continuous variables. Patients were stratified into two surgical groups: BCS and mastectomy. Categorical variables were compared between the two groups using Chi-square tests (χ^2^) for variables with more than two categories or Fisher’s exact tests for binary variables. To identify predictors of surgical treatment, univariate analyses were conducted for all pre-operative variables, followed by multivariate logistic regression for variables with a *p*-value < 0.05 in the univariate analysis. The logistic regression analysis provided odds ratios (OR) and 95% confidence intervals (95% CI) to determine independent factors influencing the choice of surgery in BC patients treated with NAT. Long-term oncologic outcomes were assessed, including DFS, DDFS, OS, and BCSS. DFS was defined as the time from the date of BC surgery to the first occurrence of tumor recurrence (either loco-regional or distant). DDFS was defined as the time interval between BC surgery and the detection of distant metastases. OS was defined as the time from the treatment of BC to death for any cause or at the last known follow-up. BCSS was defined as the time from surgery to death specifically attributed to BC. The Kaplan–Meier method was employed to estimate recurrence and survival probabilities for the two surgical groups, and differences between groups were evaluated using the log-rank test. To identify independent risk and protective factors for oncologic outcomes, Cox proportional hazards regression models were performed, with hazard ratios (HRs) and 95% CIs reported. The last follow-up was updated to 27 December 2024. Statistical significance was set at a two-sided *p*-value < 0.05. Data analyses and visualizations were conducted using IBM SPSS Statistics version 25.0.

## 3. Results

### 3.1. Baseline Characteristics of the Study Population

A total of 607 BC patients who underwent NAT were included in the analysis. The median age of the patients was 51 years (range, 20–88), with 339 (55.9%) being post-menopausal. Pre-operative imaging was comprehensive, with 457 patients (75.3%) undergoing mammography, all 607 patients (100%) undergoing breast and axillary US, 278 patients (45.8%) having axillary biopsy, 347 patients (57.2%) undergoing breast MRI, and 456 patients (75.1%) receiving PET. The median tumor size prior to NAT was 30 mm (range, 7–100 mm). Clinical staging at diagnosis revealed that 127 patients (20.9%) were cT1, 370 (61.0%) were cT2, 66 (10.9%) were cT3, and 44 (7.2%) were cT4. Additionally, 175 patients (28.8%) were clinically node-negative (cN0), while 432 (71.2%) presented with nodal involvement (cN+). Regarding NAT regimens, targeted therapy consisted of trastuzumab in 267 patients (44.0%), pertuzumab in 18 (3.0%), and pembrolizumab in 21 (3.5%). The most common histologic subtype was ductal carcinoma (*n* = 565, 93.1%), and the majority of patients presented with a single tumor nodule (*n* = 452, 74.5%). Overall, pCR was achieved in 153 patients (25.2%), and the median tumor size post-NAT was reduced to 6 mm (range, 0–100 mm). Details of baseline and post-NAT characteristics are summarized in Table 1.

### 3.2. Comparison of Characteristics Between Surgical Groups

Of the 607 patients, 332 (54.7%) underwent BCS, and 275 (45.3%) underwent mastectomy. Multivariate analysis identified independent predictors for BCS. cT1-2 stage at diagnosis significantly increased the likelihood of BCS (OR: 2.966, 95% CI: 1.751–5.022, *p* < 0.001), as did cN0 status at diagnosis (OR: 1.820, 95% CI: 1.234–2.684, *p* = 0.003). The presence of a single nodule was also associated with a higher probability of BCS (OR: 0.478, 95% CI: 0.322–0.708, *p* < 0.001). The results of multivariate analyses are detailed in Table 2.

### 3.3. Long-Term Oncological Outcomes

After a median follow-up of 54 months (range, 12–194 months), patients treated with BCS showed significantly higher survival and lower recurrence rates across all metrics compared to those who underwent mastectomy. The 10-year DFS rates were 75.2% for BCS versus 71.1% for mastectomy (*p* = 0.001). Similarly, the 10-year DDFS rates were 75.2% for BCS compared to 71.1% for mastectomy (*p* = 0.001). The OS rates at 10 years were 82.9% for BCS and 78.1% for mastectomy (*p* = 0.002), and the BCSS rates were 87.7% and 83.1%, respectively (*p* = 0.001). These findings are summarized in Table 3 and illustrated with Kaplan–Meier survival curves (Figure 2 and Figure 3).

### 3.4. Independent Prognostic Factors for Recurrence and Survival

Cox regression analyses identified key prognostic factors for recurrence and survival outcomes. Achieving pCR after NAT significantly improved DFS (HR: 0.335, 95% CI: 0.147–0.763, *p* = 0.009), DDFS (HR: 0.371, 95% CI: 0.162–0.847, *p* = 0.019), OS (HR: 0.168, 95% CI: 0.044–0.468, *p* = 0.010), and BCSS (HR: 0.170, 95% CI: 0.032–0.890, *p* = 0.036). Adjuvant radiotherapy was a predictor of better DFS (HR: 0.581, 95% CI: 0.346–0.975, *p* = 0.040). Similarly, adjuvant endocrine therapy was a predictor of better DFS (HR: 0.463, 95% CI: 0.248–0.865, *p* = 0.016) and DDFS (HR: 0.492, 95% CI: 0.263–0.920, *p* = 0.026). Additionally, T-DM1 was a strong predictor of better OS (HR: 0.341, 95% CI: 0.185–0.629, *p* = 0.001) and BCSS (HR: 0.262, 95% CI: 0.119–0.576, *p* = 0.001). Conversely, vascular invasion was associated with poorer outcomes, including decreased OS (HR: 1.963, 95% CI: 1.136–3.394, *p* = 0.016), and mastectomy was associated with lower BCSS (HR: 2.068, 95% CI: 1.016–4.210, *p* = 0.045). Independent factors influencing survival and recurrence outcomes are detailed in Table 4.

Since adjuvant radiotherapy was identified as a predictor of better DFS, the study population of BC patients treated with NAT was further stratified into three subgroups based on both surgical and radiotherapy treatment. Given that all patients who underwent BCS received adjuvant radiotherapy, the following three subgroups were created: BCS versus mastectomy with adjuvant radiotherapy (Ms+RT) versus mastectomy without adjuvant radiotherapy (Ms−RT). Among the 275 patients treated with mastectomy, 158 (57.5%) received adjuvant radiotherapy, while 117 (42.5%) did not. When comparing oncologic outcomes across the three subgroups, BCS remained associated with superior survival and recurrence outcomes. Patients undergoing BCS demonstrated significantly higher DFS and DDFS rates compared to both Ms+RT and Ms−RT groups (*p* = 0.004). Similarly, BCS was associated with superior OS (*p* = 0.007) and BCSS (*p* = 0.002) compared to both mastectomy subgroups (Figure 4 and Figure 5).

## 4. Discussion

Our study demonstrated that BCS following NAT is oncologically safe and associated with comparable, if not superior, long-term outcomes when compared to mastectomy. The rationale for conducting this study arises from the ongoing uncertainties regarding the comparative oncological safety of BCS versus mastectomy in BC patients undergoing NAT. These uncertainties have persisted despite major advancements in surgical techniques, imaging, and systemic therapies, largely because recent large-scale meta-analyses have reported conflicting results. One key meta-analysis, performed by the EBCTCG, included 4756 women across ten randomized trials comparing NAT with adjuvant chemotherapy [30]. While the study confirmed that NAT was associated with a higher probability of BCS (65% in the NAT group versus 49% in the adjuvant group, *p* < 0.0001), it also showed a significantly higher 15-year local recurrence rate in NAT-treated patients (21.4% versus 15.9%, rate ratio: 1.37, 95% CI: 1.17–1.61, *p* = 0.0001). Importantly, there were no significant differences in the rate of distant recurrence (38.2% versus 38.0%, rate ratio: 1.02, 95% CI: 0.92–1.14, *p* = 0.66) or BC-specific mortality (34.4% versus 33.7%, rate ratio: 1.06, 95% CI: 0.95–1.18, *p* = 0.31). However, this meta-analysis had several limitations, including the inclusion of outdated treatment regimens, a lack of detailed data on radiotherapy, and variable local therapy practices among the included trials (i.e., two studies did not conduct surgical intervention post-NAT in cases of clinical complete response), raising concerns about the applicability of these findings in modern clinical settings. A second meta-analysis by Qin et al. [31] analyzed ten comparative studies encompassing 5018 patients, of whom 2120 underwent BCS and 2898 underwent mastectomy post-NAT. This study reported superior 5-year OS (OR: 2.68, 95% CI: 2.19–3.28, *p* < 0.00001) and DFS (OR: 3.11, 95% CI: 1.80–5.38, *p* < 0.0001) in the mastectomy group compared to the BCS group. However, the conclusions that could be drawn were significantly limited by heterogeneity in patient populations, limited follow-up duration, variability of treatment plans, and the lack of randomization in some of the included trials, which introduced a potential for selection bias. Overall, these meta-analyses indicated a need for further research regarding the safety and efficacy of BCS as compared to mastectomy in the NAT setting, considering new developments in systemic and local therapies.

Recent evidence supports the oncological safety and even superiority of BCS over mastectomy in patients with BC undergoing NAT, particularly when significant tumor downstaging is achieved. A meta-analysis by Sun et al. [25], including 16 studies with a pooled cohort of 3531 patients, demonstrated that BCS following NAT was associated with significantly reduced distant recurrence rates (OR: 0.51, 95% CI: 0.42–0.63, *p* < 0.01) and improved DFS (OR: 2.35, 95% CI: 1.84–3.01, *p* < 0.01) and OS (OR: 2.12, 95% CI: 1.51–2.98, *p* < 0.01). In particular, BCS was associated with a 49% reduced risk of distant recurrence and more than double the DFS and OS rates compared to mastectomy, highlighting the procedure’s feasibility in patients achieving substantial tumor downstaging. Similarly, the New Jersey State Cancer Registry analysis performed by Arlow et al. [26] showed that BCS followed by adjuvant radiotherapy resulted in significantly improved 10-year BCSS compared to mastectomy without radiotherapy (*p* = 0.0046). Even after propensity score-matching analyses to account for baseline differences between treatment groups, the superiority of BCS + radiotherapy was evident (HR: 0.46, 95% CI: 0.27–0.78). In another large-scale study, Nobrega et al. [34] analyzed a cohort of 530 patients with locally advanced BC; of these, 24.6% underwent BCS, while 75.4% underwent mastectomy. After a median follow-up of 79 months, superior 6-year OS rates in the BCS group (81.5%) versus the mastectomy group (62%, *p* < 0.001) were reported. Mastectomy remained a significant predictor of worse OS (OR: 1.67, 95% CI: 1.06–2.63, *p* = 0.024). Conversely, achieving a pCR after NAT, was strongly associated with improved OS (OR: 0.42, 95% CI: 0.22–0.80, *p* = 0.008). In an extensive analysis of a Korean cohort consisting of 1641 patients treated with NAT, Gwark et al. [35] corroborated these findings, showing that BCS combined with adjuvant radiotherapy yielded superior DFS (HR: 0.45, 95% CI: 0.36–0.57, *p* < 0.0001), DDFS (HR: 0.41, 95% CI: 0.32–0.53, *p* < 0.0001), and OS (HR: 0.39, 95% CI: 0.30–0.51, *p* < 0.0001) compared to mastectomy. This advantage persisted even in patients with advanced nodal disease, emphasizing the broader applicability of BCS in cases where nodal involvement may initially suggest more aggressive surgical management. Similarly, in our study, adjuvant radiotherapy was identified as a significant predictor of improved DFS, further supporting the protective role of radiotherapy in achieving favorable oncologic outcomes. Additional studies further substantiate these findings. Kim et al. [36] analyzed 45,770 patients with BC from the Korean Breast Cancer Registry and found that BCS with radiotherapy consistently demonstrated superior OS and BCSS compared to mastectomy, both before and after exact matching of prognostic factors. Chu et al. [37] evaluated 214,128 women with BC from the SEER registry and demonstrated that BCS with radiotherapy yielded better 5-year OS and cause-specific survival than mastectomy across all molecular subtypes. Even after adjusting for demographic and clinical factors, the risk of mortality remained significantly higher for mastectomy compared to BCS + radiotherapy. Lastly, Krug et al. [38] analyzed 2632 patients from the German Breast Group meta-database, demonstrating that in patients with early-stage cN0 triple-negative BC treated with NAT, BCS was not associated with increased loco-regional recurrence compared to mastectomy. In fact, after a median follow-up of 64 months, BCS was associated with significantly better DFS (HR: 0.51, *p* < 0.001) and OS (HR: 0.43, *p* < 0.001). Absence of pCR was identified as the strongest predictor of locoregional recurrence (HR: 2.22, *p* = 0.001) and worse DFS (HR: 2.43, *p* < 0.001) and OS (HR: 3.15, *p* < 0.001).

Additionally, in the field of reconstructive surgery, Bogdan et al. [39] have highlighted the critical role of advanced assessment techniques in evaluating outcomes, including fat graft volume retention. Techniques such as three-dimensional scanning, MRI, and US provide non-invasive, accurate methods for quantifying changes in graft volume. These methodologies can potentially enhance the precision of reconstructive procedures and improve patient outcomes, especially in breast reconstruction after oncologic surgery.

Our study has several limitations. First, its retrospective design introduces potential biases, including selection bias in surgical decision-making and variability in the application of adjuvant therapies. Second, although data were retrieved from a single high-volume center, limiting heterogeneity, it may also constrain the generalizability of findings to other institutions with differing patient populations, treatment protocols, or resource availability. Third, while our Cox regression analysis did not identify tumor stage (T and N) as an independent prognostic factor for recurrence and survival, there was a statistically significant difference in clinical T/N stage between the BCS and mastectomy groups at multivariate analysis, as shown in Table 2. In fact, while our findings support the oncologic safety of BCS in appropriately selected cases, they do not suggest that mastectomy is inherently inferior to BCS as a surgical choice. As a matter of fact, a prospective study with case-matching by stage would be required to confirm these conclusions. Finally, although the follow-up period was long enough to assess long-term outcomes, further prospective studies with larger sample sizes and longer follow-up periods are required to confirm these results and further assess the changing role of systemic therapies in optimizing surgical outcomes. Additionally, socioeconomic factors and accessibility to follow-up care, while indirectly accounted for by the uniform treatment protocols and structured follow-up at our institution, were not specifically analyzed in this study.

## 5. Conclusions

In conclusion, our findings reveal that BCS, when feasible, achieves superior long-term oncologic outcomes compared to mastectomy in BC patients treated with NAT, including higher DFS, DDFS, OS, and BCSS. These results demonstrate the oncologic safety and potential superiority of BCS in appropriately selected patients, particularly those achieving significant tumor downstaging and favorable pathologic responses. This highlights the importance of individualized surgical decision-making to optimize survival outcomes.

## Figures and Tables

**Figure 1 cancers-17-00766-f001:**
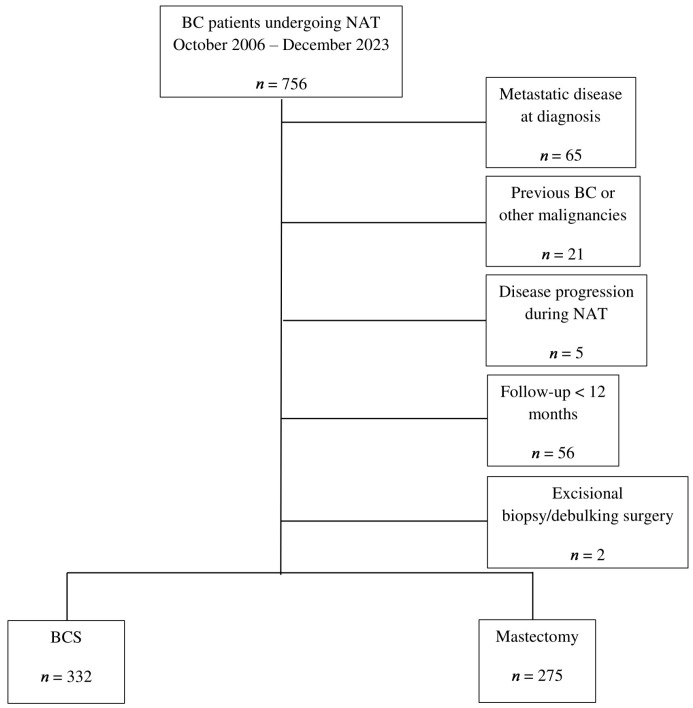
Flowchart illustrating the inclusion and exclusion criteria for the study population, starting with 756 breast cancer (BC) patients treated with neoadjuvant therapy (NAT) and resulting in the final cohort of 607 patients categorized into BCS (*n* = 332) and mastectomy (*n* = 275) groups.

**Figure 2 cancers-17-00766-f002:**
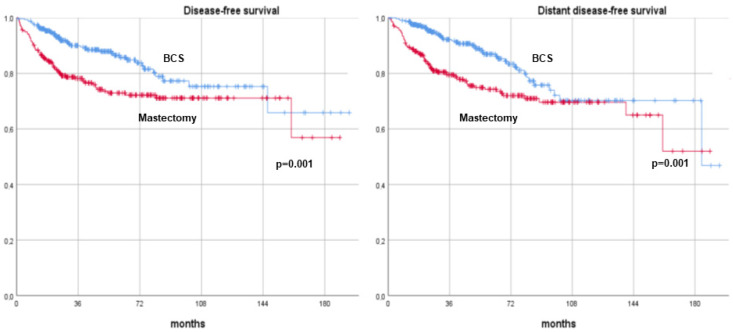
Disease-free survival and distant disease-free survival curves of breast cancer patients undergoing neoadjuvant therapy and surgery (breast-conserving surgery versus mastectomy). BCS: breast-conserving surgery.

**Figure 3 cancers-17-00766-f003:**
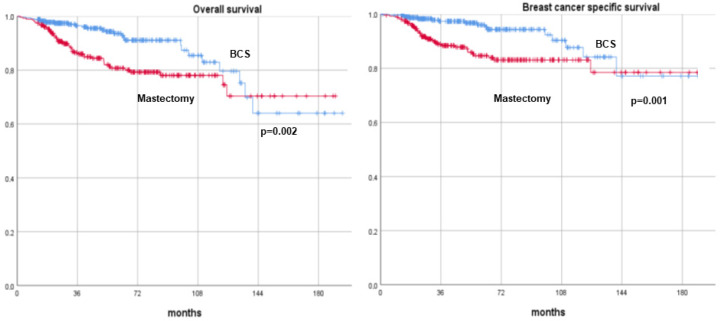
Overallsurvival and breast cancer-specific survival curves of breast cancer patients undergoing neoadjuvant therapy and surgery (breast-conserving surgery versus mastectomy). BCS: breast-conserving surgery.

**Figure 4 cancers-17-00766-f004:**
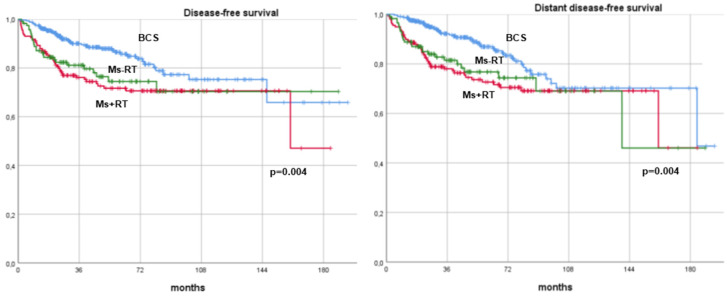
Disease-free survival and distant disease-free survival curves of breast cancer patients undergoing neoadjuvant therapy. Patients were stratified based on their surgical approach (breast-conserving surgery versus mastectomy) and further divided according to receipt of adjuvant radiotherapy. BCS: breast-conserving surgery, Ms+RT: mastectomy with adjuvant radiotherapy, Ms−RT: mastectomy without adjuvant radiotherapy.

**Figure 5 cancers-17-00766-f005:**
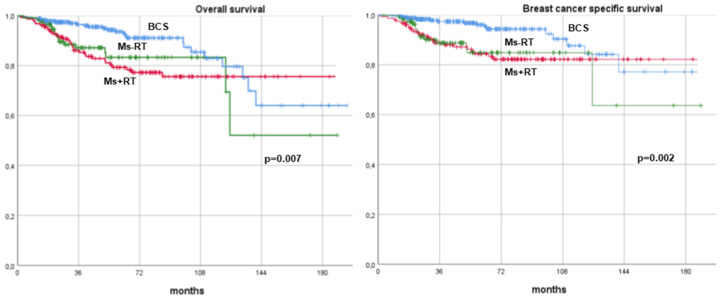
Overallsurvival and breast cancer-specific survival curves of breast cancer patients undergoing neoadjuvant therapy. Patients were stratified based on their surgical approach (breast-conserving surgery versus mastectomy) and further divided according to receipt of adjuvant radiotherapy. BCS: breast-conserving surgery, Ms+RT: mastectomy with adjuvant radiotherapy, Ms−RT: mastectomy without adjuvant radiotherapy.

**Table 1 cancers-17-00766-t001:** Baseline characteristics of 607 breast cancer patients undergoing neoadjuvant therapy.

Characteristics	Number (%)/Median (Range)
**Demographics**	
Age (years)	51(20–88)
Post-menopausal	339 (55.9%)
**Pre-operative staging**	
Mammography	457 (75.3%)
Breast and axillary US	607 (100%)
Axillary biopsy	278 (45.8%)
MRI	347 (57.2%)
PET	456 (75.1%)
Dimension pre-NAT (mm)	30 (7–100)
Stage pre-NAT	
cT1	127 (20.9%)
cT2	370 (61.0%)
cT3	66 (10.9%)
cT4	44 (7.2%)
cN0	175 (28.8%)
cN+	432 (71.2%)
**Neoadjuvant therapy**	
NAT with anthracycline only	43 (7.1%)
NAT without anthracycline	20 (3.3%)
NAT with anthracycline and taxanes	543 (89.5%)
Trastuzumab	267 (44.0%)
Pertuzumab	18 (3.0%)
Pembrolizumab	21 (3.5%)
**Tumor**	
Subtype	
Luminal-like	171 (28.2%)
HER2-positive	266 (43.8%)
Triple-negative	170 (28.0%)
Histotype	
Ductal	565 (93.1%)
Lobular	21 (3.5%)
Other	21 (3.5%)
Vascular invasion	114 (18.8%)
Single nodule	452 (74.5%)
**Pathologic response**	
pCR	153 (25.2%)
Dimension post-NAT (mm)	6 (0–100)
Stage post-NAT	
ypT0	168 (27.7%)
ypTis	61 (10.1%)
ypTmi	14 (2.3%)
ypT1a	46 (7.6%)
ypT1b	77 (12.7%)
ypT1c	115 (19.0%)
ypT2	98 (16.1%)
ypT3	20 (3.3%)
ypT4	8 (1.2%)
ypN0	391 (64.4%)
ypNi+	3 (0.5%)
ypNmi	24 (4.0%)
ypN1a	95 (15.7%)
ypN2a	61 (10.1%)
ypN3	33 (5.3%)
**Surgical treatment**	
BCS	332 (54.7%)
Mastectomy	275 (45.3%)
SLNB not followed by ALND	345 (56.8%)
SLNB followed by ALND	87 (14.3%)
Direct ALND	175 (28.9%)
**Post-operative treatment**	
Taxanes	37 (6.1%)
Capecitabine	60 (9.9%)
Radiotherapy	483 (79.6%)
Endocrine	307 (50.6%)
T-DM1	186 (30.6%)
Abemaciclib	16 (2.6%)

Footnotes: US: ultrasound, MRI: magnetic resonance imaging, PET: positron emission tomography, NAT: neoadjuvant therapy, HER2: HER2 evaluated either on immunohistochemistry or on in situ hybridization, according to the ASCO CAP guidelines, pCR: pathologic complete response, BCS: breast-conserving surgery, SLNB: sentinel lymph node biopsy, ALND: axillary lymph node dissection, T-DM1: trastuzumab emtansine.

**Table 2 cancers-17-00766-t002:** Predictors of surgical treatment type in breast cancer patients undergoing neoadjuvant therapy.

Characteristics	BCS (*n* = 332) Tot. (%)	Mastectomy (*n* = 275) Tot. (%)	Multivariate Analysis *p*-Value OR (95% CI)
**Demographics**			
Age (years)			
≤51	158(47.6%)	166 (60.4%)	0.120 0.674 (0.410–1.108)
>51	174 (52.4%)	109 (39.6%)	-
Menopausal status			
Pre-menopausal	132 (39.8%)	136 (49.5%)	0.350 0.789 (0.480–1.297)
Post-menopausal	200 (60.2%)	139 (50.5%)	-
**Pre-operative staging**			
Dimension pre-NAT (mm)			
≤30	230 (69.3%)	141 (51.3%)	0.281 1.248 (0.834–1.866)
>30	102 (30.7%)	134 (48.7%)	-
Stage pre-NAT			
cT1-2	298 (89.8%)	199 (72.4%)	<0.001 ^a^ 2.966 (1.751–5.022)
cT3-4	34 (10.2%)	76 (27.6%)	-
cN0	117 (35.2%)	58 (21.1%)	0.003 ^a^ 1.820 (1.234–2.684)
cN+	215 (64.8%)	217 (78.9%)	-
**Tumor**			
Subtype			
Luminal-like	78 (23.5%)	93 (33.8%)	0.565 0.935 (0.744–1.175)
HER2-positive	162 (48.8%)	104 (37.8%)	-
Triple-negative	92 (27.7%)	78 (28.4%)	-
Single nodule			
Yes	270 (81.3%)	182 (66.2%)	<0.001 ^a^ 0.478 (0.322–0.708)
No	62 (18.7%)	93 (33.8%)	-

Footnotes: BCS: breast-conserving surgery, OR: odds ratio, 95% CI: 95% confidence interval, NAT: neoadjuvant therapy, ^a^: statistically significant.

**Table 3 cancers-17-00766-t003:** Comparison of disease-free, distant disease-free, overall survival, and breast cancer-specific survival in breast cancer patients undergoing neoadjuvant therapy.

Outcomes	BCS	Mastectomy	*p*-Value
DFS rate			0.001 ^a^
3-year	90.0%	78.2%	
5-year	86.5%	72.9%	
10-year	75.2%	71.1%	
DDFS rate			0.001 ^a^
3-year	92.2%	79.5%	
5-year	86.9%	74.2%	
10-year	75.2%	71.1%	
OS rate			
3-year	96.5%	86.5%	
5-year	93.6%	80.7%	0.002 ^a^
10-year	82.9%	78.1%	
BCSS			
3-year	97.4%	88.9%	
5-year	96.1%	84.6%	0.001 ^a^
10-year	87.7%	83.1%	

Footnotes: DFS: disease-free survival, DDFS: distant disease-free survival, OS: overall survival, BCSS: breast cancer specific survival, ^a^: statistically significant.

**Table 4 cancers-17-00766-t004:** Independent Prognostic Factors for Recurrence and Survival Outcomes in Breast Cancer Patients Undergoing Neoadjuvant Therapy.

Independent Factors	DFS HR (95% CI) *p*-Value	DDFS HR (95% CI) *p*-Value	OS HR (95% CI) *p*-Value	BCSS HR (95% CI) *p*-Value
**Patient**				
Age (years)				
≤51	Reference	Reference	Reference	Reference
>51	1.266 (0.697–2.300) 0.439	1.345 (0.737–2.455) 0.334	2.138 (1.002–4.563) 0.050	1.365 (0.573–3.251) 0.483
Menopausal status				
Pre-menopausal	Reference	Reference	Reference	Reference
Post-menopausal	0.707 (0.390–1.282) 0.254	0.715 (0.392–1.302) 0.272	0.887 (0.410–1.918) 0.761	1.000 (0.417–2.401) 1.000
**Pre-operative staging**				
Dimension pre-NAT (mm)				
≤30	Reference	Reference	Reference	Reference
>30	0.868 (0.536–1.404) 0.563	0.851 (0.528–1.370) 0.506	0.843 (0.472–1.504) 0.562	0.706 (0.350–1.423) 0.330
Single nodule				
No	Reference	Reference	Reference	Reference
Yes	0.909 (0.581–1.423) 0.675	0.870 (0.555–1.365) 0.545	0.795 (0.457–1.383) 0.417	0.992 (0.508–1.936) 0.981
Stage pre-NAT				
cT1-2	Reference	Reference	Reference	Reference
cT3-4	1.730 (0.992–3.017) 0.053	1.618 (0.928–2.823) 0.090	1.005 (0.493–2.049) 0.989	1.443 (0.628–3.314) 0.388
cN0	Reference	Reference	Reference	Reference
cN+	0.898 (0.548–1.473) 0.671	0.917 (0.556–1.512) 0.733	0.984 (0.498–1.943) 0.962	1.088 (0.490–2.413) 0.863
**Tumor**				
Histotype				
Ductal	Reference	Reference	Reference	Reference
Other	0.954 (0.501–1.815) 0.886	0.978 (0.518–1.845) 0.945	0.817 (0.404–1.649) 0.572	1.108 (0.469–2.616) 0.816
Subtype				
Luminal-like	Reference	Reference	Reference	Reference
HER2-positive	1.031 (0.692–1.535) 0.883	1.058 (0.708–1.579) 0.784	1.600 (0.911–2.811) 0.102	1.560 (0.765–3.182) 0.221
Triple-negative				
Dimension post-NAT (mm)				
≤6	Reference	Reference	Reference	Reference
>6	1.298 (0.801–2.101) 0.290	1.362 (0.840–2.208) 0.210	1.058 (0.570–1.965) 0.858	1.846 (0.829–4.112) 0.134
Stage post-NAT				
no pCR	Reference	Reference	Reference	Reference
pCR	0.335 (0.147–0.763) 0.009 ^a^	0.371 (0.162–0.847) 0.019 ^a^	0.168 (0.044–0.468) 0.010 ^a^	0.170 (0.032–0.890) 0.036 ^a^
ypN0	Reference	Reference	Reference	Reference
ypN+	1.441 (0.766–2.710) 0.257	1.456 (0.770–2.752) 0.247	1.442 (0.642–3.237) 0.375	1.346 (0.544–3.328) 0.520
Vascular invasion				
No	Reference	Reference	Reference	Reference
Yes	1.266 (0.795–2.015) 0.321	1.230 (0.773–1.957) 0.382	1.963 (1.136–3.394) 0.016 ^a^	1.482 (0.787–2.788) 0.223
**Treatment**				
Operation				
BCS	Reference	Reference	Reference	Reference
Mastectomy	1.272 (0.801–2.019) 0.308	1.291 (0.813–2.051) 0.279	1.720 (0.962–3.074) 0.067	2.068 (1.016–4.210) 0.045 ^a^
no ALND	Reference	Reference	Reference	Reference
ALND	1.715 (0.912–3.223) 0.094	1.569 (0.828–2.973) 0.167	2.148 (0.903–5.110) 0.084	2.101 (0.789–5.594) 0.138
Adjuvant radiotherapy				
No	Reference	Reference	Reference	Reference
Yes	0.581 (0.346–0.975) 0.040 ^a^	0.604 (0.361–1.012) 0.056	0.638 (0.340–1.198) 0.162	0.535 (0.258–1.110) 0.093
Capecitabine				
No	Reference	Reference	Reference	Reference
Yes	1.335 (0.813–2.193) 0.254	1.502 (0.910–2.480) 0.112	1.070 (0.577–1.985) 0.830	1.096 (0.538–2.233) 0.801
Endocrine therapy				
No	Reference	Reference	Reference	Reference
Yes	0.463 (0.248–0.865) 0.016 ^a^	0.492 (0.263–0.920) 0.026 ^a^	0.689 (0.267–1.778) 0.441	0.575 (0.166–1.992) 0.383
T-DM1				
No	Reference	Reference	Reference	Reference
Yes	0.819 (0.537–1.247) 0.351	0.845 (0.554–1.289) 0.434	0.341 (0.185–0.629) 0.001 ^a^	0.262 (0.119–0.576) 0.001 ^a^

Footnotes: DFS: disease-free survival, DDFS: distant disease-free survival, OS: overall survival, BCSS: breast cancer-specific survival, HR: hazard ratio, 95% CI: 95% confidence interval, NAT: neoadjuvant therapy, HER2: HER2 evaluated either on immunohistochemistry or on in situ hybridization, according to the ASCO CAP guidelines, pCR: pathologic complete response, BCS: breast-conserving surgery, ALND: axillary lymph node dissection, T-DM1: trastuzumab emtansine, ^a^: statistically significant.

## Data Availability

Data supporting reported results can be found in the Appendix A.

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
