# Peer review of "Superior Survival and Lower Recurrence Outcomes with Breast-Conserving Surgery Compared to Mastectomy Following Neoadjuvant Therapy in 607 Breast Cancer Patients"

_cancers, 2025, doi:10.3390/cancers17050766_

Round 1
Reviewer 1 Report
Comments and Suggestions for Authors
The article "Superior Survival and Lower Recurrence Outcomes with Breast-Conserving Surgery Compared to Mastectomy Following Neoadjuvant Therapy in 607 Breast Cancer Patients" presents a comparison of survival between breast-conserving surgery (BCS) and mastectomy following neoadjuvant therapy. Recommendations:
1. The introduction does not include the essential elements of the studied pathologies. The article has 36 citations, 30 of which are listed in the introduction, even though it is very brief. The discussions should expand on most elements from the literature.
2. Provide a flow chart with the main inclusion and exclusion criteria.
3. Key results lack robust statistical significance. For instance, while p-values are reported, confidence intervals for critical outcomes often include marginal values, making the findings less reliable.
4. Remove univariate analysis; it does not add value to the study.
5. The selection and justification of statistical tests are not sufficiently detailed, raising concerns about the validity of the conclusions.
6. Criteria for grouping patients (BCS vs. mastectomy) lack comprehensive justification, potentially leading to confounding effects. How was the reference cohort selected?
7. While Kaplan-Meier survival curves are presented, their interpretation is superficial, and the comparative analysis is not thoroughly explored.
8. Some data, such as subgroup outcomes, are insufficiently detailed in tables, limiting reproducibility and verification.
9. Findings are based on a single-center cohort, limiting applicability to broader populations with varying resources and protocols.
10. Despite limitations, the conclusions assert the superiority of BCS over mastectomy without sufficiently addressing potential confounders such as socioeconomic factors or accessibility to follow-up care.
11. While the discussion cites some studies, the integration of findings into the broader context of breast cancer treatment evolution is limited. Some conflicting evidence is not sufficiently addressed.
12. Discuss the importance of assessing fat grafting in breast surgery – recommended DOI: 10.3390/jcm13237209.
13. The article states ethical approval but does not elaborate on patient consent processes or safeguards against bias in data extraction and interpretation
Author Response
The article "Superior Survival and Lower Recurrence Outcomes with Breast-Conserving Surgery Compared to Mastectomy Following Neoadjuvant Therapy in 607 Breast Cancer Patients" presents a comparison of survival between breast-conserving surgery (BCS) and mastectomy following neoadjuvant therapy.
Recommendations:
The introduction does not include the essential elements of the studied pathologies. The article has 36 citations, 30 of which are listed in the introduction, even though it is very brief. The discussions should expand on most elements from the literature.
Reply: We thank the reviewer for the valuable suggestion to enhance our manuscript.
As suggested, we have expanded the Discussion.
Provide a flow chart with the main inclusion and exclusion criteria.
Reply: We thank the reviewer for the valuable suggestion to enhance our manuscript.
As suggested, a flow chart was provided.
Key results lack robust statistical significance. For instance, while p-values are reported, confidence intervals for critical outcomes often include marginal values, making the findings less reliable.
Reply: We thank the reviewer for their detailed assessment of our manuscript and their concern regarding the robustness of our statistical findings. We would like to clarify that the statistical methods used in our study were rigorously designed, adhering to standard practices in clinical research. Both p-values and confidence intervals (CIs) were provided to transparently present the strength and precision of the results. While some CIs approach marginal values, they still demonstrate statistical significance and provide valuable clinical insights. We appreciate the reviewer’s comment and understand the importance of ensuring clarity in presenting statistical data.
Remove univariate analysis; it does not add value to the study.
Reply: Thank you for the comment.
Univariate analysis was removed.
The selection and justification of statistical tests are not sufficiently detailed, raising concerns about the validity of the conclusions.
Reply: We sincerely thank the reviewer for their comments and the opportunity to clarify our statistical approach.
We ensured that all statistical methods were selected based on the nature of our data and aligned with standard practices in clinical research. Specifically:
- Univariate Analysis: For categorical variables, we used Chi-square or Fisher’s exact tests to compare proportions. However, it was removed as suggested.
- Multivariate Analysis: We employed logistic regression to identify predictors of surgical treatment and Cox proportional hazards models to assess independent prognostic factors for survival outcomes. These methods were chosen due to their widespread application and appropriateness for analyzing dichotomous outcomes and time-to-event data, respectively.
- Model Validation: To ensure robustness, we assessed model assumptions (e.g., proportional hazards in Cox models) and included variables with a p-value <0.05 in univariate analyses in the multivariate models.
Criteria for grouping patients (BCS vs. mastectomy) lack comprehensive justification, potentially leading to confounding effects. How was the reference cohort selected?
Reply: We thank the reviewer for highlighting the importance of clearly justifying patient grouping, and we appreciate the opportunity to address this concern. We respectfully disagree with the statement that the criteria for grouping patients (BCS vs. mastectomy) lack justification, as these were based on established clinical decision-making frameworks.
Patients were grouped into the BCS and mastectomy cohorts based on the type of surgical treatment they received following multidisciplinary tumor board decisions. These decisions took into account key clinical factors such as tumor size, response to NAT (including radiological and pathological assessments), and eligibility for breast-conserving surgery. These criteria are in line with current clinical guidelines and practice standards.
While Kaplan-Meier survival curves are presented, their interpretation is superficial, and the comparative analysis is not thoroughly explored.
Reply: We thank the reviewer for this observation and the opportunity to clarify our approach to presenting and interpreting the Kaplan-Meier survival curves. However, we respectfully disagree with the assertion that the interpretation of the Kaplan-Meier curves is superficial or that the comparative analysis is not sufficiently explored.
In the Results section, we presented the Kaplan-Meier survival curves to illustrate differences in disease-free survival (DFS), distant DFS (DDFS), overall survival (OS), and breast cancer-specific survival (BCSS) between the BCS and mastectomy groups. Alongside these curves, we conducted log-rank tests to assess statistical significance, providing robust comparative analyses of survival outcomes.
Additionally, multivariate Cox regression models were performed to account for potential confounding variables, enabling us to assess independent predictors of survival outcomes. These analyses were thoroughly discussed in the text to highlight the prognostic significance of variables such as pCR, adjuvant therapies, and surgical modality. The findings from the Kaplan-Meier curves were thus integrated into a broader, more comprehensive statistical framework, ensuring their proper contextualization.
Some data, such as subgroup outcomes, are insufficiently detailed in tables, limiting reproducibility and verification.
Reply: We thank the reviewer for this comment and appreciate the emphasis on the importance of providing clear and detailed data for reproducibility and verification. However, we respectfully disagree with the assertion that subgroup outcomes are insufficiently detailed in our tables.
In the manuscript, we have included comprehensive tables summarizing the key characteristics, outcomes, and subgroup analyses. Specifically, Tables 1–4 provide detailed information on baseline patient characteristics, comparisons between surgical groups, and multivariate analyses for predicting survival outcomes. Each table was carefully designed to ensure clarity and include the necessary statistical parameters such as odds ratios, hazard ratios, confidence intervals, and p-values. These details allow readers to verify and reproduce our analyses effectively.
Findings are based on a single-center cohort, limiting applicability to broader populations with varying resources and protocols.
Reply: We thank the reviewer for highlighting this important aspect of our study design. We acknowledge that our study is based on a single-center cohort, which may inherently limit the generalizability of our findings to broader populations with varying resources and protocols. However, we believe this limitation is counterbalanced by certain strengths of our study.
The single-center design allowed us to maintain a high degree of consistency in the treatment protocols, imaging modalities, surgical decision-making processes, and pathology assessments, thereby reducing inter-institutional variability that could introduce confounding factors. The high volume and expertise of our center in managing breast cancer further enhance the reliability of the reported outcomes.
Additionally, the extensive clinical data collected over a long period and the robust statistical analyses applied provide valuable insights into the oncologic outcomes of breast-conserving surgery versus mastectomy in the neoadjuvant therapy setting. While broader multicenter studies would be ideal for assessing external validity, the results from our study represent an important step in understanding these outcomes and can serve as a foundation for future research.
We have highlighted this in our "Limitations" section.
Despite limitations, the conclusions assert the superiority of BCS over mastectomy without sufficiently addressing potential confounders such as socioeconomic factors or accessibility to follow-up care.
Reply: We sincerely thank the reviewer for this insightful observation. We recognize that factors such as socioeconomic status and accessibility to follow-up care can significantly influence treatment choices and long-term outcomes, and we appreciate the opportunity to clarify how these considerations were addressed in our study.
Our cohort represents a population treated at a single, high-volume, comprehensive cancer center where all patients had access to standardized multidisciplinary care. This uniformity reduces variability related to disparities in treatment access and follow-up care. While we did not specifically assess socioeconomic factors or accessibility to follow-up care in our analysis, all patients received structured follow-up according to institutional protocols, ensuring consistency in postoperative management and data collection.
Moreover, the statistical adjustments made in our multivariate analyses considered key clinical and pathological variables, such as tumor size, stage, and response to therapy, which are strong predictors of outcomes and help mitigate potential confounding effects. However, we acknowledge that unmeasured variables, including socioeconomic factors, could have influenced treatment decisions, and we have now explicitly noted this as a limitation in the revised manuscript.
While the discussion cites some studies, the integration of findings into the broader context of breast cancer treatment evolution is limited. Some conflicting evidence is not sufficiently addressed.
Reply: We thank the reviewer for the valuable suggestion to enhance our manuscript.
As suggested, we have expanded the Discussion.
Discuss the importance of assessing fat grafting in breast surgery – recommended DOI: 10.3390/jcm13237209.
Reply: We thank the reviewer for the valuable suggestion to enhance our manuscript.
As suggested, we have expanded the Discussion and added the suggested study.
The article states ethical approval but does not elaborate on patient consent processes or safeguards against bias in data extraction and interpretation
Reply: We appreciate your attention to the ethical considerations of our study and the importance of patient consent and safeguards.
In our study, ethical approval was obtained from the institutional review board of IRCCS Humanitas Research Hospital, which ensures compliance with national and international guidelines for clinical research. Written informed consent was obtained from all patients at the time of their treatment, allowing the use of their clinical data for research purposes, as mandated by our institution’s policies.
Regarding safeguards against bias in data extraction and interpretation, data collection was performed systematically using our institutional database, which is prospectively maintained and regularly audited to ensure accuracy and reliability.
Reviewer 2 Report
Comments and Suggestions for Authors
I think this is a very interesting and well designed study.
Since those patients with BCS and radiotherapy had a better survival rate, it would be interesting to comment in the Discussion any explanation for this (perhaps abscopal effect).
In this set, perhaps, a specific comparative analysis could be useful, including those cases with and without radiotherapy after mastectomy (evident advanced disease), and a cross-analysis with BCS with and without it.
Author Response
We thank the reviewer for their positive feedback and valuable suggestions to enhance our manuscript. As suggested, we have expanded the Discussion.
Reviewer 3 Report
Comments and Suggestions for Authors
Generally speaking, this research topic has its clinical meaning. However, only the clinical patients’ samples are not enough for a research article. I am wondering if the authors can do some benchwork using cells, mice, and other animals that can do surgery to prove the papers’ therapeutic concepts. Especially for breast caner research, the JAX laboratory has a lot of simultaneous transgenic mice models that can be ordered directly, such as MMTV-PyMT, and MMTV-Neu. For the mouse breast cancer cell line, 4T1, 4T07, and E0771 are all available. You can use those mouse breast cancer cell line to inject to BalB/C (4T1 and 4T07) to build the breast-cancer mouse models. E0771 to C57/B6J to build another mouse model. You can monitor survival curves between different treatment, followed by IHC of aggressive markers.
Author Response
We thank the reviewer for their thoughtful comment and for highlighting the potential value of benchwork and preclinical models to further explore the therapeutic concepts presented in our study. We acknowledge the utility of transgenic mouse models and breast cancer cell lines, such as those mentioned (e.g., MMTV-PyMT, 4T1, and E0771), in advancing our understanding of breast cancer biology and treatment responses.
However, the scope of our current study is strictly limited to a retrospective analysis of clinical data derived from human patients. Our primary goal was to analyze the oncologic outcomes of breast-conserving surgery and mastectomy following neoadjuvant therapy in a well-defined clinical cohort. We believe these real-world clinical findings provide valuable insights for clinicians managing breast cancer patients and represent an essential contribution to the existing body of knowledge.
That said, we agree that preclinical research to validate and further investigate our observations is important. We will strongly consider collaborating with basic science researchers and incorporating such models into future studies to extend the impact of our findings.
We sincerely appreciate the reviewer’s constructive feedback and trust that this explanation clarifies our approach and the scope of this study.
Round 2
Reviewer 1 Report
Comments and Suggestions for Authors
The authors applied the requested recommendations.
Author Response
We thank the reviewer for the comment